# Non-Thermal Plasma Decontamination Using a Multi-Hollow Surface Dielectric Barrier Discharge: Impact of Food Matrix Composition on Bactericidal Efficacy

**DOI:** 10.3390/foods12020386

**Published:** 2023-01-13

**Authors:** Klaas De Baerdemaeker, Amber Van Reepingen, Anton Nikiforov, Bruno De Meulenaer, Nathalie De Geyter, Frank Devlieghere

**Affiliations:** 1Research Unit Food Microbiology and Food Preservation (FMFP), Department of Food Technology, Safety and Health, Ghent University, Coupure Links 653, 9000 Ghent, Belgium; 2Research Unit Plasma Technology (RUPT), Department of Applied Physics, Ghent University, Sint-Pietersnieuwstraat 41, 9000 Ghent, Belgium; 3NutriFOODchem Research Group, Department of Food Technology, Safety and Health, Ghent University, Coupure Links 653, 9000 Ghent, Belgium

**Keywords:** multi-hollow SDBD, cold plasma, non-thermal technologies, food matrix, bacterial inactivation, lipid oxidation

## Abstract

The non-thermal plasma (NTP) treatment of food products as an alternative for thermal processing has been investigated over the last few years. This quasi-neutral gas contains a wide variety of reactive oxygen and nitrogen species (RONS), which could be lethal for bacterial cells present in the product. However, apart from only targeting bacteria, the RONS will also interact with components present in the food matrix. Therefore, these food components will protect the microorganisms, and the NTP treatment efficiency will decrease. This effect was investigated by supplementing a plain agar medium with various representative food matrix components. After inoculation with *Escherichia coli* O157:H7 (STEC) MB3885, the plates were treated for 30 s by a multi-hollow surface dielectric barrier discharge (MSDBD) generated in either dry air or air at 75% humidity, at constant power (25.7 ± 1.7 W). Subsequently, the survival of the cells was quantified. It has been found that the addition of casein hydrolysate (7.1 ± 0.2 m%), starch (2.0 m%), or soybean oil (4.6 m%) decreased the inactivation effect significantly. Food products containing these biomolecules might therefore need a more severe NTP treatment. Additionally, with increasing humidity of the plasma input gas, ozone levels decreased, and the bactericidal effect was generally less pronounced.

## 1. Introduction

Thermal treatment of food products for preservation purposes has been used for around 6000 years, initially by drying and smoking of the foods. Later, with the industrial revolution, thermal pasteurization and sterilization technologies were developed, increasing the preservation power even more [1]. However, in recent times, the consumer has shown an increased interest in minimally processed foods that have the characteristics of freshness but without compromising on safety [2]. As conventional thermal treatment processes result in the loss of heat-sensitive nutritional components and changes in texture and organoleptic qualities, non-thermal techniques have been developed to respond to consumer demand [3,4]. Technologies such as high-pressure processing, pulsed electric field, and irradiation are non-thermal and established concepts in the food industry, but in the last years, a growing interest is also shown in non-thermal plasma (NTP) for bacterial inactivation [5]. PlAgri (2022) reported that the annual production of scientific documents in the field of plasma technology in the food industry increased from one to 255 in the period from 2003 to 2020 [6]. Surowsky et al., (2015) even mentioned around 800 publications dealing with non-thermal plasma-based microbial inactivation in foods in 2013 [7].

The concept of plasma was first described by Langmuir (1928) to define the gas region containing balanced charges of ions and electrons [8]. Plasma reactivity is the result of the presence of free electrons and radicals, ions, excited atoms and molecules, reactive oxygen and nitrogen species (RONS), and electromagnetic radiation (UV photons and visible light) [9,10,11,12]. Although the electrons in a non-thermal plasma are at a temperature of 10^4^–10^5^ °C, the heavy particles are close to ambient temperature. Apart from avoiding heat degradation, the advantages of NTP for food treatment are its energy efficiency [13], the possibility to operate at atmospheric pressure (so avoiding the need for vacuum equipment) [14], and the fact that it could be created from gasses that are conventionally used for modified atmosphere packaging (MAP) [15]. Additionally, for the treatment of solid foods, NTP acts only on the surface of the product, retaining the nutritional qualities on the inside [16].

The bactericidal effect of NTP treatment is the result of various modes of action. NTP irradiation is said to cause the denaturation of membrane proteins, being detrimental to the survival and duplication of the cell [17]. Reactive oxygen species will oxidize and/or damage essential biomolecules such as DNA, proteins, enzymes, lipids, and fatty acids (e.g., in the cell membrane). The cell membrane and cell wall will be disintegrated by chemical alterations and the breaking of important bonds, including C-O, C-N, and C-C. This will cause cell leakage and loss of cell functionality [18]. On the other hand, there is no scientific consensus on the contribution of UV photons in NTP to bacterial decontamination as they show bactericidal potential in various ways but are easily absorbed by the gas atoms and molecules at atmospheric pressure [19,20].

Earlier research by De Baerdemaeker et al., (2022) and Huang et al., (2020) showed that the bactericidal effect of NTP decreased when bacteria were inoculated and treated on real food products compared to agar plates. This could be explained by, among other things, the roughness of the food surface, protecting the bacterial cells [21,22]. Han et al., (2020) showed that this negative correlation between surface roughness and bacterial inactivation was linear [23]. Furthermore, Ziuzina et al., (2015) reported the entrance of bacterial cells into the pores of plant leaves, reducing the antimicrobial efficacy of NTP [24]. However, as food constituents such as lipids and proteins are prone to oxidation by ROS [25,26], it would be expected that those components also affect NTP’s efficacy. Nevertheless, there are only limited studies on the effect of the food matrix. In the current study, the aim is to explore this matrix effect by investigating the impact of various food components on bacterial inactivation using non-thermal plasmas generated from the air at both low and high relative humidity (RH) based on the humidity of gasses recommended for food packaging [27] and the composition of the plasma.

## 2. Materials and Methods

First, agar media supplemented with various food components at different levels representative of their potential presence in food products were prepared. The plates were inoculated with a standardized amount of *Escherichia coli* cells, and the NTP was treated at constant power (25.7 ± 1.7 W) and using air at 0% or 75% relative humidity (high levels and near absence of ozone, respectively, see Section 3.1), after which the bacterial cells were recovered, and surviving cells were quantified. A comparison of the recovery between the different components, their concentrations, and the humidity of the input gas allowed for understanding the effect of those components and relative humidity on bactericidal efficacy. Additionally, the plasma was characterized under the same conditions, and NTP inactivation on different supplemented media was evaluated with respect to the plasma composition.

### 2.1. Supplemented Agar Media Preparation

Agar plates were prepared, containing plain (pure) bacteriological agar (15 g/L, LP0011B, Oxoid), which was supplemented with one of the three concentrations of a certain major food matrix component (Table 1), each agar containing only one type of biomolecule. For all those components, changes in inactivation levels with increasing concentration were compared against a reference sample with no addition of the particular component but containing all chemicals facilitating its dissolution and undergoing the same preparation methods. For low-humidity plasma, all concentrations (reference, low, middle, and high) were analyzed, while the reference sample was only compared with the highest concentration when applying high-humidity plasma. Seven different components from five classes were added to the agars: proteins (casein hydrolysate (22090, Sigma-Aldrich, St. Louis, MO, USA)), carbohydrates (glucose (CL00.0710, Chem-Lab, Zedelgem, Belgium) and soluble starch (S9765, Sigma-Aldrich)), lipids (refined soybean oil and stripped soybean oil (AH slaolie, Albert Heijn, Zaandam, The Netherlands)), salt (NaCl (CL00.1429, Chem-Lab)), and anti-oxidants (β-carotene (C9750, Sigma-Aldrich)). Casein is a protein with very little secondary and tertiary structure [28], meaning all the amino acids are available, and none are structurally protected from interaction with the RONS. With the hydrolyzed protein, also referred to as peptone from casein, this is even more the case. Soybean oil consists of a good mixture of saturated fatty acids (SFA, 15%), mono-unsaturated fatty acids (MUFA, 23%), and poly-unsaturated fatty acids (PUFA, 62%) [29,30]. The refined oil was prepared by removing only oxidation products already present in commercial soybean oil by means of column chromatography on silica gel (1.15101, Millipore, Burlington, MA, USA). The stripped oil was, in addition, stripped of naturally occurring anti-oxidants (column chromatography with aluminum oxide (11503, Thermo Fisher Scientific, Waltham, MA, USA) following the silica gel column chromatography). Column chromatography was carried out using the method described by Wang et al., (2018) [31].

For the glucose and NaCl, the procedure for medium preparation was identical. Plain agar medium and a solution of the food constituent, both at double concentration, were autoclaved and filter sterilized (0.45 µm pore size, 296–4545, Thermo Fisher Scientific, Waltham, MA, USA), respectively, and subsequently aseptically mixed at a 1:1 ratio. A casein medium was prepared by making a casein hydrolysate solution at a double concentration in distilled water and adapting the pH to 10.5 by the addition of a 0.1 M and 10 M NaOH (71690, Sigma-Aldrich) solution. This casein solution was sonicated for 30 min at 50 °C and filter sterilized (0.2 µm pore size, 596–4520, Thermo Fisher Scientific, Waltham, MA, USA), and the loss of precipitate was quantified by analyzing the weight of the filter after drying. The sterile solution was mixed with plain double-strength agar medium (1:1), and the pH was reduced to 8.5 in a sterile manner with a 1 M and 6 M HCl (30721, Sigma-Aldrich) solution. For the starch agar medium, plain agar medium and a concentrated starch solution were mixed and sterilized (autoclaved). Soybean oil, whether stripped or unstripped of anti-oxidants, was mixed with Tween 20 (233362500, Acros Organics, Morris Plains, NJ, USA) while heating and kept at 48 °C before the addition of a sterile agar medium of higher strength to form a homogeneous mixture. The anti-oxidant β-carotene was first suspended in a mixture of dimethyl sulfoxide (DMSO, CL00.0422, Sigma-Aldrich) and Tween 20 and subsequently added to a sterile agar medium. To ensure an equal concentration of DMSO (5.8 mL/100 g medium) and Tween 20 (1.2 mL/100 g medium) over all supplemented agar media, an extra amount of the mixture of those components (without β-carotene) was required for some agar media and the reference.

Since the distance between the sample and electrodes has been shown to affect NTP bactericidal potential (data not shown), the height and, therefore, the volume of (supplemented) agar medium in all plates was standardized to ensure equal treatment. For all different samples, small Petri dishes (Ø 5.5 cm) were filled with 10 mL of the respective supplemented agar medium. Plates with soybean oil or β-carotene were stored for a maximum of 1 week in an anaerobic environment (AnaeroGen, AN0025A, Thermo Fisher Scientific, Waltham, MA, USA) to prevent oxidation of the biomolecules. All plates were kept at 4 °C until further use.

### 2.2. Plasma Treatment

Plasma was generated using the multi-hollow surface dielectric barrier discharge (MSDBD) setup shown in Figure 1. The MSDBD consists of two gold mesh electrodes placed at a distance of 0.20 mm with the interelectrode space filled with a dielectric barrier (Al_2_O_3_). An airflow (synthetic air, 14746, Air Products), whether or not humidified to 75% RH (near absence of ozone, see Section 3.1) by bubbling through a water column of ca. 89 cm, of 5 standard L/min (slm) went perpendicular through 260 holes (Ø 1.00 mm), arranged symmetrically in a hexagonal configuration in the electrode system. This gas flow was controlled by mass flow controllers (Bronkhorst). The discharge was generated by a high voltage pulse generator (Redline Technologies) at alternating current (frequency of 64.01 kHz) and supplied energy to the system at constant power (25.7 ± 1.7 W). Samples were NTP treated for 30 s at a distance of 18 mm from the electrodes, after which the active atmosphere was removed by flushing the reactor for 2 more minutes before opening.

Under those conditions, the temperature increase of the plasma exhaust, measured with an iButton data logger (Maxim Integrated), was limited to ca. 8.5 °C, which proved the non-thermal properties of the applied NTP. The plasma-treated gas was analyzed by Fourier-transformed infrared spectroscopy (FTIR) using a Matrix-MG2 Bruker FTIR spectrometer, enabling the quantitative analysis of the concentrations of NO, NO_2_, N_2_O_5_, N_2_O, and ozone. The detection was performed by a collection of the effluent gas and stable products at a 3 m distance from the plasma reactor. The optical multi-pass gas cell of 5.0 m length was used to measure the absorption, and the absolute calibrations were performed by Bruker. Spectra were obtained with an average of 50 scans with a resolution of 0.5 cm^−1^. The system was flushed thoroughly with air for at least 15 min in between measurements.

### 2.3. Strain Preparation

*Escherichia coli* O157:H7 (STEC) MB3885 was obtained from the FMFP (Ghent University) culture collection as cryobeads in a 15% glycerol (CL00.0736, Chem-Lab) in brain heart infusion (BHI, CM1135, Oxoid) broth at −75 °C. Two beads were transferred to fresh BHI and incubated for 2 days at 37 °C. A pure culture, obtained by the four-quadrant streak method (37 °C, 1 day) on tryptone soy agar (TSA, CM0131, Oxoid), was transferred to TSA slants (37 °C, 1 day) and kept as such for no more than 6 weeks (4 °C).

The day before every experiment, a loopful of the subculture was transferred to fresh BHI broth and incubated at 37 °C for ca. 1 day. One mL of the homogenized cell suspension was brought in an Eppendorf tube and centrifuged (8000 rpm, 5 min), after which the supernatant was replaced by fresh peptone physiological solution (PPS). This centrifugation and washing step was repeated twice more, resulting in an inoculum at a high concentration. Finally, the cell suspension was diluted to obtain an inoculum at a concentration of ca. 7 log CFU/mL. Actual concentrations and purity were determined by the standard pour-plating technique with both TSA and RAPID’E.coli 2 Medium (3564024, Bio-Rad Laboratories Inc., Hercules, CA, USA).

### 2.4. Sample Handling

The (supplemented) media plates were inoculated with 0.1 mL (ca. 6 log CFU/sample) of the diluted inoculum and dried for 15 min by air. Consequently, the plates were NTP treated (see Section 2.2). Cells were recovered from the (supplemented) agar medium by making a tenfold dilution with PPS and extraction in a Stomacher device. Survival of STEC was quantified by the standard pour-plating technique with TSA.

### 2.5. Statistical Analysis

All tests regarding inactivation levels on agar media supplemented with different concentrations of the food components were performed in triplicates. Inactivation was determined as the difference in survival of treated and their respective untreated plates. Homoscedasticity was determined by Levene’s test (based on the median) with α = 0.01. For plasma generated from dry air (comparison of four concentration levels from the same component), ANOVA (post-hoc Tukey or Games-Howell in case of homo- or heteroscedasticity, respectively) was used for analyzing the significance (α = 0.05) of the differences in inactivation levels. For humid air plasma (comparison of two concentration levels from the same component), the same comparison was made by means of the independent sample *t*-test. For both levels of RH, the inactivation on agars supplemented with monomeric glucose was compared with the inactivation on starchy (polymeric glucose) agars at the same concentration by using the independent sample *t*-test (α = 0.05). The same statistical tests were performed to compare both types of oily agars. Finally, for every component at a certain concentration, the effect of relative humidity of the plasma input gas was analyzed by the independent sample *t*-test (α = 0.05).

## 3. Results

Plain agar plates were supplemented with various food matrix components in different concentrations, inoculated with STEC, and treated with non-thermal plasma in a multi-hollow surface dielectric barrier discharge. Plasma was generated from either dry air (0% RH) or humid air (75% RH).

### 3.1. Plasma Characterization

The concentrations of various long-living RONS (N_2_O, N_2_O_5_, NO, NO_2_ and O_3_) generated during the first 50 s of treatment were measured for input air with increasing RH using Fourier-transform infrared spectroscopy. These characterization results are shown in Figure 2. With increasing %RH, the ozone concentration in the plasma drops remarkably, from 796 ± 1 ppm at 0% RH to 9 ± 1 ppm at 100% RH. On the other hand, the concentrations of NO and even more so NO_2_ increase with increasing %RH. N_2_O levels remain quasi-constant, while N_2_O_5_ decreases with increasing RH and is already absent at 75% RH. These decreasing ozone and N_2_O_5_ concentrations and increasing levels of NO_2_ and NO at increasing RH will have an impact on the antibacterial efficiency of the MSDBD treatment. This motivates the choice to compare the bactericidal potential of plasma at 0% and 75% RH with an ozone-rich atmosphere and the near absence of ozone in dry and humid air plasma, respectively. Next to this, an RH of 75% is representative of the relative humidity of the headspace of packaged microbiologically unstable food products.

### 3.2. NTP Inactivation with Dry and Humid Air Plasma

Inactivation levels of STEC when using dry air for plasma generation are shown in Figure 3. Without the addition of food components, inactivation reached levels of around 3.8 log CFU/sample, except for β-carotene (2.7 log CFU/sample, possibly due to the reaction between DMSO used for making a stable agar media and OH radicals in the plasma [32]). However, all those reference media reached the limit of detection (LOD) for at least one of three repeats (with the same exception), so inactivation could, in reality, be even higher. Agar medium supplemented with casein did show a reduced bactericidal effect with increasing protein concentration, although this was only significant (*p* ≤ 0.05) for the highest level of casein addition (7.1 ± 0.2 m%). For starch agar, the decrease in inactivation potential was even more explicit, and even at the lowest concentration (2.0 m%) evaluated, the effect of starch on the inactivation of STEC was statistically significant (*p* ≤ 0.05). Middle and high-starch concentrations did not differ significantly from each other. In contrast to the starch polysaccharide, its monomeric building blocks (glucose) did not significantly alter the NTP inactivation of STEC (*p* > 0.05), even at concentrations as high as 18.0 m%. This resulted in a significant difference (*p* ≤ 0.05) between inactivation on high-starch and high-glucose agar media. Additionally, STEC inactivation in the presence of lipids differed significantly from the reference sample, and this was for all investigated concentrations (low, middle, and high). However, for all tested concentrations (except the middle one), there was no significant difference between stripped and refined oil. Finally, in analogy to media supplemented with glucose, the addition of β-carotene or NaCl did not result in any significant changes in bactericidal effect (*p* > 0.05), although for the latter component, a slight but not statistically relevant decrease in inactivation was noticed.

Levels of inactivation for plasma generated from humid air at 75% RH are shown as red bars in Figure 4. A significant effect (*p* ≤ 0.05) of casein addition was observed, even resulting in a complete drop of bactericidal potential to approximately 0 log CFU/sample. Just as for dry air plasma, it was statistically shown that the polysaccharide starch chain negatively affected the decontamination efficiency (*p* ≤ 0.05), while this was not the case for the monomeric glucose units (*p* > 0.05). Furthermore, the effect of refined oil and stripped oil was quasi-identical: a decrease from >3.0 log CFU to ca. 1.0 log CFU inactivation after the addition of the lipid component. Finally, neither β-carotene nor NaCl impacted the humid air plasma inactivation.

For non-thermal plasma treatment on unsupplemented (reference) media, STEC inactivation was always lower when humidified air was used for plasma generation (see Figure 4). However, this was only significant (*p* ≤ 0.05) for the casein-, glucose- and NaCl-reference media. Although these food matrix components themselves were not present in the reference media, the additional chemicals used during media preparation (see Table 1) and the sometimes rather high standard deviation might have played a role. When the food constituents were added in high concentrations, only for the casein- and β-carotene-supplemented media did the NTP inactivation differ significantly at 75% RH compared to 0% RH.

## 4. Discussion

The main long-living RONS were characterized using FTIR spectroscopy. As shown in Figure 2, the most abundant long-living RONS in the dry air MSDBD exhaust was ozone. Its production in non-thermal air plasma happens by various pathways, although the mechanism is rather complex due to the presence of nitrogen [13,33]. The main reaction for ozone formation involves O_2_, atomic oxygen (O), and molecular oxygen or nitrogen (M). O was formed due to electron impact dissociation of molecular O_2_. This simple mechanism, given by Equations (1) and (2), does not include reactions with excited species or other molecules (e.g., NO and NO_2_), nor does it take into account ozone self-destruction reactions [33].
e + O_2_ → O + O + e,(1)
O + O_2_ + M → O_3_ + M,(2)

The ozone concentration drops drastically with the increasing humidity of the carrier gas. According to Patil et al., (2014), this could be attributed to the quenching/attachment of electrons to H_2_O, leading to water dissociation and a direct ozone reaction with the water molecules. It has been shown by optical absorption spectroscopy (OAS) that increasing %RH results in a lower ozone concentration due to the formation of N_2_O_5_, peroxides (mainly H_2_O_2_), HNO_4_, OH and to a lesser extent also N_2_O_4_ and HONO [34]. Nevertheless, in the current study, the concentration of N_2_O_5_ decreased (Figure 2), while the other RONS mentioned were not measured due to the limitation of the IR detection of the short-living RONS. It is said that the electronic dissociation of water in a non-thermal plasma leads to the formation of an OH radical, which rapidly reacts with ozone to form the peroxy-radical HO_2_. The latter component itself also reacts with ozone molecules, again forming OH and O_2_ [13]. Correspondingly, it is expected that the chemistry initiated by NTP in conditions of dry air and humid air should be substantially different. In the case of dry air, the main mechanism will be ozone driven, whereas active nitrogen species and peroxide radicals are defining the chemistry at high RH.

The bactericidal potential of non-thermal plasma technology for the treatment of (contaminated) solid (food) surfaces has been proven before [21,35,36,37]. Nevertheless, research has also shown that this microbial inactivation is lower when treating real food products compared to agar plates [38,39]. The reason for this observation can most probably not be attributed to one factor but is rather a combination of several elements. Han et al., (2020) have shown that surface roughness is negatively correlated with the NTP bacterial inactivation rate [23], and Ziuzina et al., (2015) found how bacteria could enter stomata on the produce surface, which protects them from inactivation by NTP [24]. However, it has been shown by De Baerdemaeker et al., (2022), although to a limited extent, that the composition of the food matrix also has a major influence on the bactericidal effect of non-thermal plasma [21].

The effect of proteins and their amino acids was investigated in the current study by supplementing casein hydrolysate (peptone from casein) to plain agar medium. As seen in Figure 3 and Figure 4, concentrations as high as 7.1 ± 0.2 m% had a significantly negative impact on the MSDBD plasma efficacy. Cataldo (2003) observed the interaction of ozone with some individual amino acids, mainly tryptophan, but also methionine, cystine, tyrosine, and phenylalanine [40]. According to Liu et al., (2019), these amino acids make up ca. 16.1 m% of the casein molecule [41]. These interactions between amino acids and the RONS might render the reactive plasma components unavailable for interaction with microorganisms, which explains the decrease in NTP inactivation efficacy on protein-rich media. In NTP generated from the air with high humidity, the ozone concentration drops close to zero, and NO_2_ seems to be most abundantly present (98.23 ppm). This nitrogen species, proven to be bactericidal [42], interacts strongly with tryptophan and, to a lesser extent, tyrosine [43], which might cause the bactericidal effect to disappear completely at 75% RH.

The presence of glucose did not seem to have an immediate impact on the bactericidal effect of NTP treatment, neither at low nor high concentrations. Interestingly, in contrast to its monomeric units, the polymeric starch chain does impact the bactericidal efficacy of NTP treatment negatively, even at concentrations of only 2.0 m%. This shows the interaction between the polymer and the reactive plasma species, which could be the result of various pathways. The depolymerization process results in the formation of several fragments [44,45,46], and although it has been reported that NTP treatment can also induce cross-linking of the chain and/or increase its molecular weight (MW) [46,47,48], the degree of polymerization and MW of the starch molecules generally decrease during plasma treatment, depending on the type of starch and treatment dose [46,47,48,49]. Additionally, oxidation of the starch by the RONS will also induce changes to the starch molecules, as well as the introduction of functional groups by the RONS [45,46,50]. Again, the more interaction between the RONS and starch, the lower the bactericidal effect of NTP treatment will be, illustrated by a decrease in bacterial inactivation at higher starch concentrations in Figure 3 and Figure 4. Both ozone and NO_2_, the major components in dry and humid air plasma, respectively (see Figure 2), are known to interact with the polysaccharide [51,52], expressed by a similar reduction of NTP inactivation potential on starchy surfaces for dry and humid air plasmas. Additionally, peroxynitrite (ONOO^−^) and hydrogen peroxide (H_2_O_2_), two bactericidal agents [53,54,55] that are known to be present in humid air plasma or the gas-liquid interface [34,56], might increase the decontamination efficiency of the humid air NTP treatment. As H_2_O_2_ reaction with starch is very slow or requires high temperatures [57], this oxygen species can still react with bacteria, even on starchy surfaces. However, analysis of peroxynitrite and hydrogen peroxide was not possible by means of FTIR, so no results on their concentrations in the plasma are available. A remark needs to be made on the surface properties of high-starch media plates, as these media had a more gel-like and rough surface that could protect the bacterial cells from reactive plasma species.

Lipids are known to be prone to oxidation, even when exposed only to air [58]. Characterization of NTP used in the current study indicated the presence of ozone, NO_2,_ and/or NO, all of which have been shown to impact the lipid oxidation process [26,58,59,60,61]. Due to this interaction between lipids and the RONS, the efficacy of the MSDBD treatment decreased both for low and high RH plasmas. It has been proven by Rød et al., (2012) that NTP treatment as short as five seconds could result in the formation of oxidation products [62]. Furthermore, there were generally no differences between inactivation levels when using soybean oil with or without its naturally occurring anti-oxidants, although it would be expected that those anti-oxidants would protect bacterial cells even more by capturing the RONS in the plasma. However, it would be too soon to conclude that these components intrinsically do not impact plasma inactivation, as their naturally occurring concentrations might be too low to have a considerable effect. It has been found that higher concentrations of α-tocopherol, an anti-oxidant present in soybean oil [63], do show a protective effect on bacteria during NTP treatment by scavenging ROS [64].

Analogously, no significant effect of β-carotene, another anti-oxidant, supplementation to the agar medium was observed. Since this vitamin A precursor is known to scavenge, e.g., ozone [65], a decrease in bacterial decontamination was expected. Possibly, the presence of DMSO in the agar impacts the effect of the β-carotene as the former has been shown to react with alkenes, abundantly present with β-carotene, with the formation of methyl sulfones. This reaction needs hydroxyl radicals, which could be supplied by the plasma discharge [66].

The addition of NaCl to the agar could impact NTP inactivation in different ways. Earlier research has shown that ozone and NaCl could form the bactericidal component hypochlorous acid [67,68]. On the other hand, the salt ions (presumably Cl^-^) may protect the microorganisms by increasing the solution density, resulting in a decreased movement of reactive plasma species and lower accessibility to bacterial cells [69]. Nevertheless, no significant effect on NTP efficacy was found by supplementing NaCl to the agars when using dry or humid air.

## 5. Conclusions

This study shows that various food matrix components do have an important effect on NTP inactivation efficacy against STEC. Treatment of (food) products with a significant lipid or starch content (>4.5 and 2.0 m%, respectively) will result in an important reduction of bacterial decontamination during treatment, although this also depends on saturation degree and amino acid composition, respectively. The concentration of anti-oxidants naturally present in the oil is too low to impact the NTP inactivation potential. On the other hand, protein content needs to be rather high (7.1 ± 0.2 m%) in order to have a clear effect, although the effect is more pronounced when humid air is used for plasma generation (no bacterial reduction at high casein concentration and 75% RH). For these reasons, protein-rich and sugary food products seem to be better suited to be NTP treated compared to greasy foods. This could be fruits and vegetables, although their surfaces will also be of major importance. Furthermore, NTP might be advantageous compared to other (conventional or novel) technologies for the microbial decontamination of lean meats. For fish and meat products with higher fat content, the treatment dose needs to be increased. Nevertheless, it needs to be taken into account that a higher treatment dose might also induce a more intense nutrient loss (e.g., protein degradation), although, for solid foods, this will be limited to the surface of the product. Therefore, these changes to the nutritional value due to the current MSDBD plasma treatment should be investigated as such. Additionally, the relative humidity of the headspace at the moment of plasma treatment of packaged products is a critical factor for the efficiency of the NTP treatment, especially when both NTP and packaging technologies are integrated into one system. These considerations are of utmost importance when an (MSDBD) non-thermal plasma treatment is implemented in the production chain and show the need for an appropriate design.

## Figures and Tables

**Figure 1 foods-12-00386-f001:**
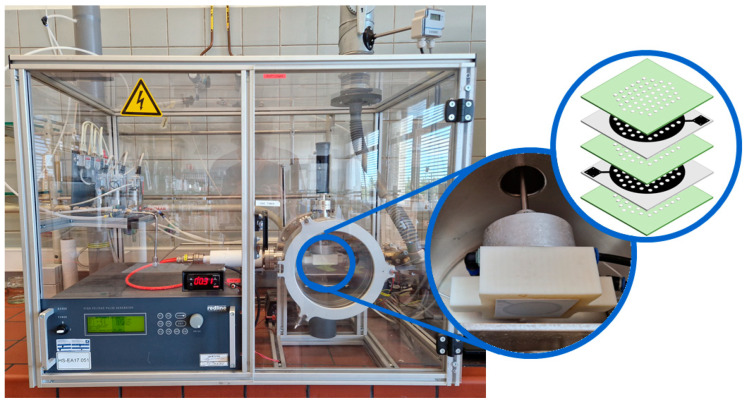
Multi-hollow surface dielectric barrier discharge setup used in the current study. On the right, a detailed image and schematic overview of the electrode system are presented (green = dielectric; black = electrode).

**Figure 2 foods-12-00386-f002:**
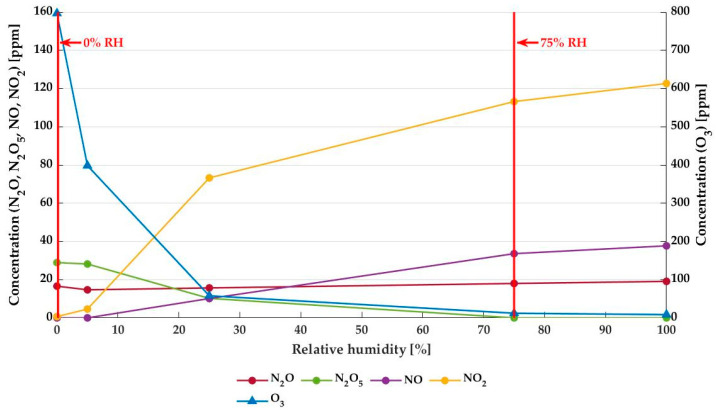
Characterization of NTP by FTIR in the first 50 s of plasma generation as a function of relative humidity of the input air. The left axis shows the concentration of N_2_O, N_2_O_5_, NO, and NO_2_ (circles); on the right axis, the concentration of ozone (O_3_) in the plasma is projected (triangles).

**Figure 3 foods-12-00386-f003:**
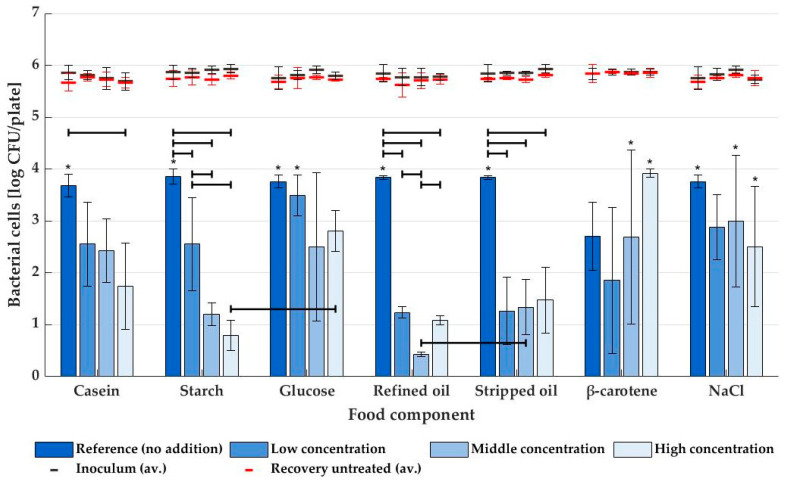
Average NTP inactivation levels (and standard deviations) of STEC on agar media supplemented with various food matrix components (*n* = 3), with plasma generated from dry air. An asterisk (*) indicates that inactivation reached LOD for at least one repeat. All components were tested at low, middle, and high concentrations (see Table 1) and compared against a reference sample without addition. The average inoculum and number of cells recovered from untreated samples are given as black and red lines, respectively. Significant differences between inactivation levels are indicated with black horizontal lines.

**Figure 4 foods-12-00386-f004:**
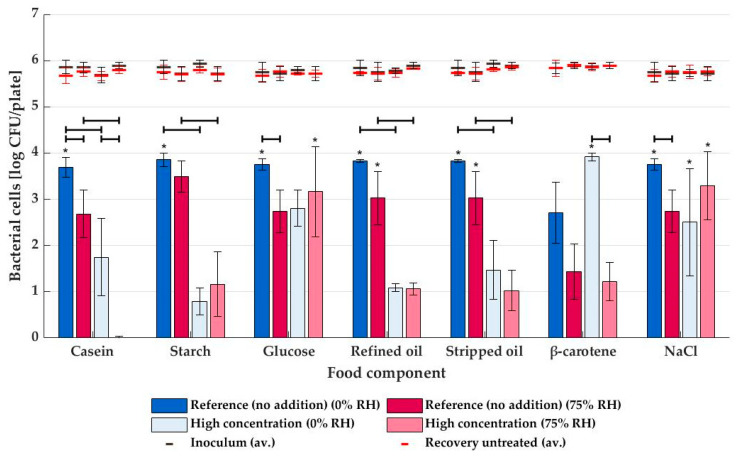
Average NTP inactivation levels (and standard deviations) of STEC on agar media supplemented with various food matrix components (*n* = 3), with plasma generated from dry air (0% RH, blue bars) and humid air (75% RH, red bars). An asterisk (*) indicates that inactivation reached LOD for at least one repeat. All components were tested at high concentrations (see Table 1) and compared against a reference sample without addition. The average inoculum and number of cells recovered from untreated samples are given as black and red lines, respectively. Significant differences between inactivation levels are indicated with black horizontal lines.

**Table 1 foods-12-00386-t001:** Concentrations of the food matrix components in the supplemented agar media applied in this study based on their presence in real-life food products. Seven different components from five classes (indicated in bold) were added to the agars.

	Low Conc. ^1^	Middle Conc. ^1^	High Conc. ^1^	Additional Chemicals ^2^
**Protein**				
Casein hydrolysate	2.6	5.1	7.1 ± 0.2	HCl and NaOH
**Carbohydrates**				
Glucose	2.0	10.0	18.0	-
Starch	2.0	10.0	18.0	-
**Lipids** ^3^				
Stripped soybean oil	4.5	21.0	37.5	Tween 20 (2.0 m%)
Refined soybean oil	4.6	21.7	38.7	Tween 20 (2.0 m%)
**Salt**				
NaCl	0.1	1.6	3.0	-
**Anti-oxidants**				
β-carotene	0.1	3.6	7.0	DMSO (5.8 *v*/*m*%) and Tween 20 (1.2 *v*/*m*%)

^1^ Concentrations are expressed as g/100 g agar medium for all components, except β-carotene (mg/100 g medium). ^2^ Present in low, medium, and high concentration agar media and in reference medium (agar medium without food component), always in same concentrations. ^3^ Concentration of triglycerides in the agar medium is identical for both types of oils, which explains the differences in addition of both types of soybean oil.

## Data Availability

The data presented in this study are openly available in Zenodo at https://doi.org/10.5281/zenodo.7385615.

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
