# Peer review of "Non-Thermal Plasma Decontamination Using a Multi-Hollow Surface Dielectric Barrier Discharge: Impact of Food Matrix Composition on Bactericidal Efficacy"

_foods, 2023, doi:10.3390/foods12020386_

Round 1

Reviewer 1 Report

Manuscript entitled „ Non-Thermal Plasma Decontamination Using a Multi-hollow Surface Dielectric Barrier Discharge: Impact of Food Matrix Composition on Bactericidal Efficacy” is methodically exploring the impact of atmospheric pressure plasma on various components present in food, when the microbial contamination is also present.

Experimental work is well planned, correlation between relative humidity and presence of reactive species is also taken under account. It is a good step towards actual application of NTP in food industry.

Generally, paper title is informative, length is suitable, quality of graphs and language are satisfactory.

Comments:

1. How do you think about penetration depth in terms of solid foods? What kind of approach (plasma source/configuration) would be the best for this case in your opinion?- I am just curious about your remarks, it does not to be included in the paper as it is quite complete.

Reviewer 2 Report

The manuscript of Klaas De Baerdemaeker et al describes the sterilisation efficiency of non-thermal plasma on different food components in different relative humidity environments. There is a clear rationale for the work and the manuscript has a logical flow. The conclusions of the manuscript are supported by the results of the experiments. However, there are some detailed doubts, so I suggest the authors revise it carefully. The manuscript requires major revisions.

Major comments:

1. Why choose 0% or 75% relative humidity air for NTP treatment?

2. Whether the nutrient content of the food product changes after NTP treatment.

3. NTP treatment of food components in agar medium is very different from real-life foods and it is recommended to supplement the results of NTP treatment of real foods, such as fish and meat products.

4. Why did the authors choose to inoculate the medium with E. coli rather than Salmonella or other bacteria in this study?

Reviewer 3 Report

This paper describes the effect of NTP on E. coli O157:H7 survival on agar plates containing various protein, lipid, and carbohydrate levels. An MSDBD apparatus was used and RONS and humidity levels were characterized and correlated with the results. The food components did have an effect on bacterial inactivation efficiency. The authors did a good job explaining their findings, which should have important implications in NTP research. 

Round 2

Reviewer 2 Report

I have no any comments for authors. I suggest that this revision can been accepted the publication.